# Association between Iron Status and Survival in Patients on Chronic Hemodialysis

**DOI:** 10.3390/nu15112577

**Published:** 2023-05-31

**Authors:** Seok-Hui Kang, Bo-Yeon Kim, Eun-Jung Son, Gui-Ok Kim, Jun-Young Do

**Affiliations:** 1Division of Nephrology, Department of Internal Medicine, Yeungnam University Medical Center, 170 Hyeonchung-Ro, Nam-Gu, Daegu 42415, Republic of Korea; kangkang@ynu.ac.kr; 2Health Insurance Review and Assessment Service, 60 Hyeoksin-Ro, Wonju-si 26465, Republic of Korea; kimby01@hira.or.kr (B.-Y.K.); conet9999@hira.or.kr (E.-J.S.); rrnlfl52@gmail.com (G.-O.K.)

**Keywords:** hemodialysis, iron, transferrin saturation, ferritin

## Abstract

The aim of this study was to evaluate survival rates according to iron status in patients undergoing maintenance hemodialysis (HD). Thus, the National HD Quality Assessment Program dataset and claims data were used for analysis (*n* = 42,390). The patients were divided into four groups according to their transferrin saturation rate and serum ferritin levels: Group 1 (*n* = 34,539, normal iron status); Group 2 (*n* = 4476, absolute iron deficiency); Group 3 (*n* = 1719, functional iron deficiency); Group 4 (*n* = 1656, high iron status). Using univariate and multivariable analyses, Group 1 outperformed the three other groups in terms of patient survival. Using univariate analysis, although Group 2 showed a favorable trend in patient survival rates compared with Groups 3 and 4, the statistical significance was weak. Group 3 exhibited similar patient survival rates to Group 4. Using multivariable Cox regression analysis, Group 2 had similar patient survival rates to Group 3. Subgroup analyses according to sex, diabetic status, hemoglobin level ≥ 10 g/dL, and serum albumin levels ≥ 3.5 g/dL indicated similar trends to those of the total cohort. However, subgroup analysis based on patients with a hemoglobin level < 10 g/dL or serum albumin levels < 3.5 g/dL showed a weak statistical significant difference compared with those with hemoglobin level ≥ 10 g/dL, or serum albumin levels ≥ 3.5 g/dL. In addition, the survival difference between Group 4 and other groups was greater in old patients than in young ones. Patients with a normal iron status had the highest survival rates. Patient survival rates were similar or differed only modestly among the groups with abnormal iron status. In addition, most subgroup analyses revealed similar trends to those according to the total cohort. However, subgroup analyses based on age, hemoglobin, or serum albumin levels showed different trends.

## 1. Introduction

Chronic kidney disease is a public health problem, attracting attention owing to the increasing aging population and chronic comorbidities. Furthermore, patients with end-stage renal disease requires renal replacement therapy. Maintenance hemodialysis (HD) is the most frequently used modality among the three renal replacement therapies. However, patients undergoing HD have been reported to have a higher mortality rate than the general population did [1]. Iron is an essential element in most human cells and a cofactor for vital proteins and enzymes [2]. Iron dysregulation is common in patients undergoing HD. Anemia in patients on HD can be caused by two important factors: insufficient erythropoietin and abnormal iron status [3]. Iron deficiency in patients with anemia could result from absolute or functional iron deficiency. Thus, a normal iron balance is essential for sustaining normal hemoglobin levels. Iron also plays an important role in maintaining mitochondria respiratory capacity in cardiomyocytes. Therefore, iron deficiency could increase morbidity and mortality in patients undergoing HD. However, the iron overload has been reported to also be associated with high mortality in patients undergoing HD. This could result from increased infection risks due to an abnormal immune response and cardiovascular complications caused by endothelial dysfunction or myocardial iron deposits [4]. Consequently, maintaining a normal iron status assists in improving patient survival.

Recent guidelines have recommended that iron status evaluations are performed at least every 3 months because of the importance of regulating iron balance in patients on HD [3]. In addition, iron and/or erythropoiesis-stimulating agents (ESAs) can be modified according to the iron status and hemoglobin levels. Previous studies have reported the importance of proper iron status [5,6,7,8,9]. Several studies have evaluated the association between transferrin saturation rate (TSAT) or ferritin levels and clinical outcomes in patients on HD, but most of the previous studies have focused on the effect on clinical outcomes of simple low-iron or high-iron status compared to those with a normal iron status as a reference group [5,6,7,8,9]. The iron status can be classified into four more detailed groups, including normal iron status, absolute and functional iron deficiencies, and iron overload. Compared to the results from simple assessments of a low- or high-iron status, analyses using four groups of iron status from a large sample may provide valuable insights into the relationship between more specific iron status and mortality. Furthermore, additional studies comparing diverse groups for iron status are needed to establish proper interventions based on the type of iron status abnormality in patients undergoing HD. The aim of this study was to evaluate patient survival according to the iron status in patients undergoing maintenance HD.

## 2. Patients and Methods

### 2.1. Study Population

This study retrospectively used data from the National HD Quality Assessment Program and the Health Insurance Review and Assessment (HIRA) of the Republic of Korea [10,11]. Briefly, HD Quality Assessment Programs were regularly conducted. The fourth and fifth programs were performed between July and December 2013 as well as between July and December 2015, respectively. Patients undergoing chronic HD were included in the programs (aged ≥18 years, ≥3 months, and ≥8 sessions per a month). We analyzed the clinical and laboratory data from the HD quality assessment program and claims data from the HIRA.

The fourth and fifth HD Quality Assessment Programs’ data (*n* = 21,846 for the fourth and *n* = 35,538 for the fifth) were analyzed. Of these, we excluded repeat participants (*n* = 13,870), patients with insufficient datasets, or those who underwent HD using a catheter (*n* = 1124). Finally, 42,390 patients were included in the study. The Yeungnam University Medical Center’s Institutional Review Board granted approval of this study (approval no. YUMC 2022-01-010), which was conducted retrospectively and thus exempted from the requirement for informed consent.

### 2.2. Study’s Variables

The collected clinical data comprised: sex, age, HD vintages (days), underlying cause of end-stage renal disease, and vascular access type. Laboratory data included: serum creatinine (mg/dL), calcium (mg/dL), albumin (g/dL), phosphorus (mg/dL), and ferritin (ng/mL); Kt/V_urea_; hemoglobin (g/dL); pre-dialysis systolic blood pressure (SBP, mmHg); pre-dialysis diastolic blood pressure (DBP, mmHg); ultrafiltration volume (UFV, percentage/body weight/session); and TSAT (%). These data were collected monthly and averaged. Despite previous guidelines, the HIRA recommends that the iron status be evaluated monthly throughout the 6 months of the HD Quality Assessment Program [3]. TSAT (%) and serum ferritin (ng/mL) levels were measured monthly over 6 months. We used the averaged data of TSAT and ferritin levels. Kt/V_urea_ was calculated using the Daugirdas’ equation [12].

The patients were divided into four groups according to TSAT and serum ferritin levels as previously described [13,14]. These include: Group 1 (normal iron status), 20% < TSAT and serum ferritin ≤ 800 ng/mL; Group 2 (absolute iron deficiency), TSAT ≤ 20% and serum ferritin ≤ 200 ng/mL; Group 3 (functional iron deficiency), TSAT ≤ 20% and serum ferritin 200–800 ng/mL; and Group 4 (high iron status), 800 ng/mL < serum ferritin. In our study, the normal iron status was defined as patients with 20% < TSAT without an upper cut-off value and serum ferritin < 800 ng/mL without a lower cut-off value. Some patients in Group 1 had 50% < TSAT with serum ferritin < 800 ng/mL or 20% < TSAT with ferritin < 100 ng/mL. These may be associated with some conditions, such as aggressive iron supplementation through IV iron or a decrease in transferrin level due to malnutrition or inflammation. However, the proportion of these patients was <5% in Group 1 and this would not have strongly influenced our results. Furthermore, all patients with 800 ng/mL < serum ferritin had 20% < TSAT.

Appendix A shows the codes of medications. In our study, iron supplementation was presented as the use of iron, which was defined as the receipt of more than one prescription regardless of the route of medication during the six-month period of each HD quality assessment. As shown in Appendix A, the iron agents included carbonyl iron, chondroitin sulfate iron, ferric chloride, ferric hydroxide polymaltose, ferritinic iron, ferrous gluconate, ferrous sulfate, ferric oxide, iron acetyl transferrin, iron dextran, iron hydroxide sucrose, iron protein succinylate, polysaccharide iron, and sodium ferric gluconate. The ESA dose (IU/week) was averaged from the total dose of ESA administered over 6 months. The ESA resistance index (ERI) was calculated using the following equation:ERI=ESA dose (IU/week)/body weight (kg)/hemoglobin (g/dL)
in the [15]. The use of anti-hypertensive drugs, aspirin, and statins as concomitant medications was also evaluated. Medication use was defined as the use of one or more prescriptions 1 year before the HD Quality Assessment Program evaluation.

One year prior to the HD Quality Assessment Program, comorbidities were assessed using the codes established by Quan et al. [16,17]. The Charlson Comorbidity Index (CCI) was used for identifying seventeen comorbidities. All patients in our study had renal disease and underwent HD. Appendix A presented the remaining comorbidities and their corresponding International Classification of Diseases-10 codes. After identifying and defining the comorbidities, the CCI score was calculated.

The follow-up was conducted until April 2022, with the endpoint being the date of transfer to peritoneal dialysis or kidney transplantation, at which point the data were censored. Electronic data were used for defining clinical outcomes during follow-up, except for death. The patients who underwent peritoneal dialysis were censored using the codes O7072, O7071, or O7061. However, those who received kidney transplants were censored using code R3280. Data on patient death were collected from the HIRA database.

### 2.3. Statistical Analyses

SAS Enterprise Guide version 7.1 (SAS Institute, Cary, NC, USA) was used to manipulate the data. Furthermore, R (version 3.5.1; R Foundation for Statistical Computing, Vienna, Austria) was used for statistical analyses. Categorical variables are shown as percentages and numbers, whilst continuous variables with a normal distribution are expressed as means and standard deviations. Continuous variables without a normal distribution are expressed as median (interquartile range). The normal distribution was evaluated using a histogram and Q–Q plot. To examine whether categorical variables were independent, Pearson’s χ^2^ test or Fisher’s exact test were used. However, a one-way analysis of variance was used for comparing the means of continuous variables with normal distribution, followed by Tukey’s post hoc test. The analysis of covariance was used to adjust for baseline characteristics. Continuous variables without normal distribution were compared using the Kruskal–Wallis test, followed by Dunn’s post hoc test with the Bonferroni correction. We used the Kaplan–Meier curves and Cox regression analyses to estimate patient survival and compared the survival curves using the log-rank test to obtain *p*-values. Multivariable analysis was performed to determine the relationships between the variables as follows: the CCI score, sex, age, type of vascular access, underlying cause of end-stage renal disease, HD vintage, UFV, Kt/V_urea_, serum calcium level, serum creatinine level, hemoglobin level, serum albumin level, serum phosphorus level, SBP, DBP, use of anti-hypertensive medications, aspirin or statins, ESA dose, ERI, and use of iron. In addition, we used categorical data based on median values for the HD vintage and ESA dose. The median values were 1229 days for HD vintage and 5130 IU/week for the ESA dose. The two variables were adjusted as dichotomous variables: the HD vintage, short group with HD < 1229 days, and long group with HD ≥ 1229 days; ESA dose, low dose with <5130 IU/week and high dose with ≥5130 IU/week. The enter mode was used for the analyses. Statistical significance was determined using a threshold of *p* < 0.05.

## 3. Results

### 3.1. Clinical Characteristics

The number of patients in Groups 1, 2, 3, and 4 were 34,539 (81.5%), 4476 (10.6%), 1719 (4.1%), and 1656 (3.9%), respectively (Table 1).

Group 1 included a greater proportion of men and arteriovenous fistulas than the other three groups did. Serum albumin levels were higher in Group 1 than in the other groups. Group 3 had a greater proportion of patients with diabetes mellitus (DM) than the other three groups did. Group 2 had a younger mean age compared with the other groups. The anti-hypertensive medication usage was the lowest in Group 2, while the aspirin or statin usage was the lowest in Group 4. Groups 3 and 4 had higher CCI scores and lower UFV, hemoglobin, phosphorus, and serum creatinine levels compared with those in Groups 1 and 2. The numbers of patients treated with iron in Groups 1, 2, 3, and 4 were 21,676 (62.8%), 2458 (54.9%), 1086 (63.2%), and 902 (54.5%), respectively (*p* < 0.001).

### 3.2. Survival Analyses

At the endpoint of follow-up, the number of patients in the survivor and death subgroups were 19,685 (57.0%) and 11,637 (33.7%) in Group 1; 2452 (54.8%) and 1630 (36.4%) in Group 2; 821 (47.8%) and 758 (44.1%) in Group 3; and 761 (46.0%) and 762 (46.0%) in Group 4, respectively (*p* < 0.001, Figure 1). The proportion of deaths at the endpoint of follow-up was the highest in Group 4. The number of patients in the peritoneal dialysis and kidney transplantation subgroups were 121 (0.4%) and 3096 (9.0%) in Group 1; 24 (0.5%) and 370 (8.3%) in Group 2; 7 (0.4%) and 133 (7.7%) in Group 3; and 11 (0.7%) and 122 (7.4%) in Group 4, respectively.

The 5-year survival rates were 69.5% in Group 1, 66.1% in Group 2, 61.0% in Group 3, and 57.5% in Group 4 (Figure 2; *p* < 0.001 for trend). The difference between Group 1 and Groups 2, 3, or 4 was significant (*p* < 0.001). Group 2 also showed a significant difference compared with Groups 3 or 4 (*p* < 0.001). However, there was no significant difference between Group 3 and Group 4 (*p* = 0.102). The Kaplan–Meier analyses revealed that, among the four group, Group 1 had the highest patient survival rate followed by Group 2. The patient survival rate did not differ significantly between Groups 3 and 4.

Compared with the other three groups, Group 1 exhibited superior patient survival rates, whereas Groups 3 and 4 had lower patient survival rates. Univariate Cox regression analysis indicated that Group 2 had a hazard ratio (HR) of 1.08 (95% confidence interval (CI), 1.03–1.13), Group 3 had an HR of 1.32 (95% CI, 1.24–1.41), and Group 4 had an HR of 1.43 (95% CI, 1.34–1.53) compared to Group 1 (Table 2).

Compared with Group 2, Group 3 had an HR of 1.22 (95% CI, 1.13–1.32) and Group 4 had an HR of 1.32 (95% CI, 1.22–1.43) (Appendix A). However, Groups 3 and 4 had comparable patient survival rates, and the multivariable Cox regression analyses indicated that Group 2 had comparable patient survival rates with those of Group 3.

We conducted subgroup analyses based on sex, age (65 years), the presence of DM, hemoglobin level (10 g/dL), and serum albumin levels (3.5 g/dL) (Figure 3).

Multivariable Cox regression analyses showed that subgroup analyses according to sex, presence of DM, hemoglobin levels ≥ 10 g/dL, and serum albumin levels ≥ 3.5 g/dL had similar trends to those of the total cohort. However, subgroup analyses based on patients with hemoglobin levels < 10 g/dL or serum albumin levels < 3.5 g/dL showed a weak statistically significant difference compared with those with a hemoglobin level ≥ 10 g/dL or serum albumin level ≥ 3.5 g/dL. In addition, the survival difference between Group 4 and the other groups was greater in patients ≥ 65 years than in those < 65 years.

### 3.3. Erythropoiesis-Related Indicators according to Iron Status

The ESA doses in Groups 1, 2, 3, and 4 were 5050 (4820), 5470 (6120), 5740 (5110), and 5510 (5130) IU/week, respectively (*p* < 0.001 for trend) The difference between Group 1 and Groups 2, 3, or 4 was significant (*p* < 0.05). Group 2 also showed a significant difference compared with Group 3 (*p* < 0.05). The ERI in Groups 1, 2, 3, and 4 was 8.0 (8.0), 8.8 (10.6), 9.4 (9.0), and 9.5 (9.7) IU/kg/g/dL, respectively (*p* < 0.001 for trend). We performed pairwise comparisons between two groups and showed that ERI was significantly lower in Group 1 than in Groups 2, 3, or 4 (*p* < 0.05). The ERI value in Group 2 was significantly lower than that in Groups 3 or 4 (*p* < 0.05). There were significant differences in baseline characteristics. Therefore, we performed multivariable analyses to identify the independent effect of the ESA dose or ERI on iron status. Multivariable analyses showed that the mean ESA doses in Groups 1, 2, 3, and 4 were 5396, 5758, 5666, and 5702 IU/week, respectively (*p* < 0.001 for trend). The mean ESA dose was significantly lower in Group 1 than in Group 2 (*p* < 0.05). The mean ERI in Groups 1, 2, 3, and 4 was 8.9, 9.7, 9.6, and 9.9 IU/kg/g/dL, respectively (*p* < 0.001 for trend). The mean ERI in Group 1 was significantly lower than those in Groups 2, 3, or 4 (*p* < 0.05).

## 4. Discussion

In the present study, we evaluated a cohort of 42,390 patients who underwent the HD Quality Assessment Program. Using both univariate and multivariable analyses, we found that Group 1 had the highest patient survival rates among the four groups. However, Group 2 had a positive trend in survival rates compared with Groups 3 and 4, albeit with weak statistical significance. These findings were consistent with those observed in subgroup analyses. Group 1 had the lowest trends in ERI and ESA doses among the four groups.

Abnormal iron balance, including deficiency and overload, has been found to be associated with high mortality in patients undergoing maintenance HD. Iron deficiency can be divided into absolute and functional categories. Iron overload shares some characteristics with functional iron deficiency, despite being different entities. Iron deficiency is associated with insufficient erythropoiesis and a high ESA dose. Consequently, these lead to highly adverse outcomes in patients undergoing HD. Recent studies have shown that hepcidin, an iron regulator, was associated with absolute and functional iron deficiency [18]. Additionally, hepcidin was associated with high rates of cardiovascular disease or mortality via direct or indirect pathways in patients undergoing dialysis [19,20,21,22]. Iron overload is associated with cellular damage via the generation of reactive oxygen species, which leads to adverse events such as premature death or atherosclerosis. Immunological dysfunction, such as decreased phagocytosis and abnormal antibody production, can develop due to iron overload [23]. Absolute, functional iron deficiency, and iron overload can be categorized as an abnormal iron status but may have different effects on clinical outcomes in patients undergoing HD. Therefore, we divided the patients into four groups according to TSAT and ferritin levels.

Previous studies demonstrated hazard effects on mortality in patients undergoing HD due to a high TSAT, high ferritin level, or iron deficiency [5,6,7,8,9]. Kalantar-Zadeh et al. evaluated 58,058 patients undergoing maintenance HD and showed that a high iron status was associated with high mortality despite the confounding effects of malnutrition–inflammation complex syndrome [5]. Pollak et al. examined 1774 patients undergoing HD and showed that those with a TSAT > 25% and ferritin > 600 had the best patient survival rates [6]. Other studies have shown an association between low TSAT levels and high mortality [7,8]. Yeh et al. evaluated patients on HD with/without polycystic kidney disease and showed that a low or high iron status was associated with high patient mortality in those without polycystic kidney disease [9]. A study of cohorts on peritoneal dialysis showed that patients with a functional iron deficiency or a high iron level had higher mortality rates than those with a normal iron level or an absolute iron deficiency did [24]. Previous studies evaluated the association between abnormal iron status and clinical outcomes; however, there are insufficient data for more diverse groups with abnormal iron status and a large sample size. Our results show the best patient survival in patients with a normal iron status and modest differences or similar patient survival among patients with an abnormal iron status.

Our study highlights several insights into iron supplementation that consider regional and ethnic issues. First, 45.1% of the patients with an absolute iron deficiency did not receive iron supplementation, while those with absolute iron deficiency underwent a greater ESA dose and ERI than did those with a normal iron status. Proper iron supplementation in Group 2 without iron supplementation and an additional iron prescription in Group 2 with iron supplementation would be useful for normalizing the iron status and decreasing ESA dosage. Consequently, this could improve patient survival. Second, 45.5% of the patients with a high iron status continued iron supplementation. Iron supplementation cessation in these patients would help avoid additional complications due to iron overload, such as cardiovascular diseases or oxidative stress. There may be several reasons for the prescription of iron supplements despite the high iron status. First, some patients in Group 4 would have been prone to an inflammatory status and had an inadequate response to ESA for managing anemia. These may lead to the prescription of iron despite high iron status. Second, iron supplements might have been prescribed without the careful consideration of a patient’s iron levels, potentially leading to indiscriminate use. Another factor that could have contributed to this issue would be the patient’s mistaken belief that iron supplements were helpful regardless of their own iron status. Although the issue is beyond the scope of our study, further studies of iron supplementation in these patients would be helpful to attenuate iron overload and its related complications.

Our subgroup analyses showed a weak association between iron status and patient survival in patients with anemia. This finding may be associated with unresolved confounding factors. In this study, patients with anemia and a normal iron status and those with iron deficiency had similar patient survival rates. Anemia in patients with a normal iron status could occur due to other factors, such as an inadequate ESA dose, hyperparathyroidism, or inflammation, which are associated with poor outcomes [3]. Non-adjustment for these factors may be associated with similar survival rates among patients with normal iron levels and patients with iron deficiency using subgroup analyses based on patients with anemia. Furthermore, the survival difference between Group 4 and the other groups was greater among old patients than among young patients. These may be associated with greater vulnerability to iron toxicity in patients ≥65 years than in those <65 years. High iron status could lead to cell damage by producing hydroxyl free radicals [25]. The older population typically experiences a decrease in defense mechanisms, particularly those related to oxidative stress, which makes them more vulnerable to the impacts resulting from high levels of iron. High mortality in Group 4 in patients ≥65 years would be associated with these issues.

Inflammation influences the TAST and the ferritin level, and these causes difficulty in distinguishing the exact iron status, thereby influencing the results. However, our study did not include the data regarding inflammation, such as C-reactive protein level and neutrophil/lymphocyte ratio, and our study did not completely exclude patients with inflammation or make an adjustment for inflammation. Nevertheless, subgroup analyses according to serum albumin levels offer partial solutions to overcome this limitation and provide valuable indications. Serum albumin is a well-known acute-phase reactant that is inversely correlated with inflammation. Normal serum albumin levels could rule out a high inflammatory condition. Findings using subgroup analyses based on patients with normal serum albumin levels were similar to those by the total cohort and may thus aid in concluding that an abnormal iron status could be an independent risk factor for mortality in patients on HD without a high inflammation status. In contrast, subgroup analyses according to patients with low serum albumin indicate that the weak association between the high iron status and patient survival would be associated with the confounding effect of inflammation or malnutrition. However, further investigations employing multivariable analysis adjusted for inflammatory indicators or conducting analyses using cohort excluding patients with inflammation, are required to establish a definitive relationship between the iron status and clinical outcomes.

Appendix A shows a summary of previous studies that evaluated the association between iron status and mortality in patients on HD using large sample sizes (*n* ≥ 10,000) [5,9,26,27,28,29]. Previous studies using large sample sizes have mainly focused on the risk of simple high ferritin levels or low iron status compared to a normal iron status, and these did not show a comparison of survival based on various abnormal iron statuses. However, iron deficiency can be classified into absolute and functional iron deficiency, and the difference in mortality between these two categories cannot be ruled out. Furthermore, these studies used the national database of DaviTa, International Dialysis Outcomes and Practice Patterns Study, or Taiwan Renal Registry Data System, and these studies were performed using a similar or limited dataset. While our research lacked significant novelty, our study has significance in its exploration of mortality associated with various abnormal iron statuses and the analysis conducted using a previously unexplored dataset concerning iron status and mortality in a new regional area.

There were certain limitations to our study. First, our study was limited by its retrospective design. Second, comorbidities and the use of medications, including iron supplementation, were only evaluated using claims data. Discrepancies between medication prescriptions and their actual use may have been present. In addition, various iron supplements could have been purchased from pharmacies without hospital prescriptions. Third, in our study, the iron status was categorized using laboratory findings during a limited duration (6 months for each HD Quality Assessment Program). In addition, our study used data from many centers, and its iron status findings may have had considerable inter-center variability. Fourth, the data analyzed in our study did not include information on the cause of death, the specific dosage of iron administered, and indicators associated with inflammation. The benefits of iron supplementation would have been observed in cardiovascular diseases or infections, and information on the cause of death would enable us to better understand the impact of an abnormal iron status beyond just mortality from any cause. In addition, there were difficulties in dosing iron supplementations. In South Korea, many iron formulas have been prescribed, and their bioavailability differs according to each patient’s characteristics, medication timing, and/or iron formulas. Therefore, we only evaluated the presence of iron prescriptions.

## 5. Conclusions

Patients who had a normal iron status exhibited the most favorable survival rates. Patient survival rates were similar or differed only modestly among the groups with an abnormal iron status. In addition, most subgroup analyses revealed similar trends to those based on the total cohort. However, subgroup analyses based on age, hemoglobin, or serum albumin levels showed a different trend. Our study findings suggest that a change in the prescription of iron towards a normal iron status would be helpful for improving the survival of patients undergoing maintenance HD.

## Figures and Tables

**Figure 1 nutrients-15-02577-f001:**
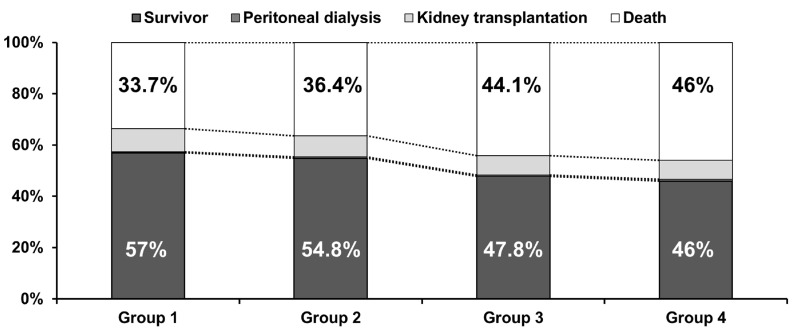
Patient status at the endpoint of follow-up.

**Figure 2 nutrients-15-02577-f002:**
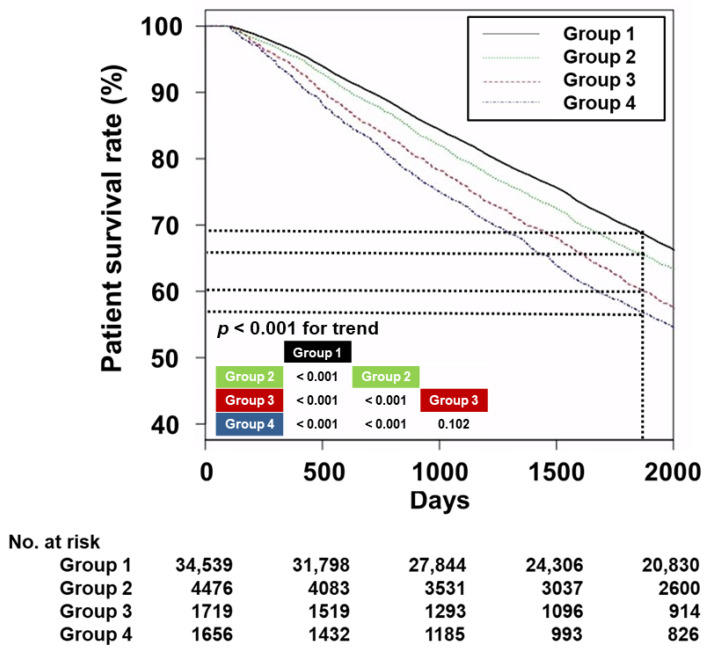
Kaplan–Meier curves depicting the survival of patients in each study group. The 5-year survival rates per group are expressed as dotted lines and *p*-values for pairwise comparison with log-rank tests added to the lower left corner of the graph. Abbreviations: Group 1, patients with normal iron status; Group 2, patients with absolute iron deficiency; Group 3, patients with functional iron deficiency; Group 4, patients with high iron status.

**Figure 3 nutrients-15-02577-f003:**
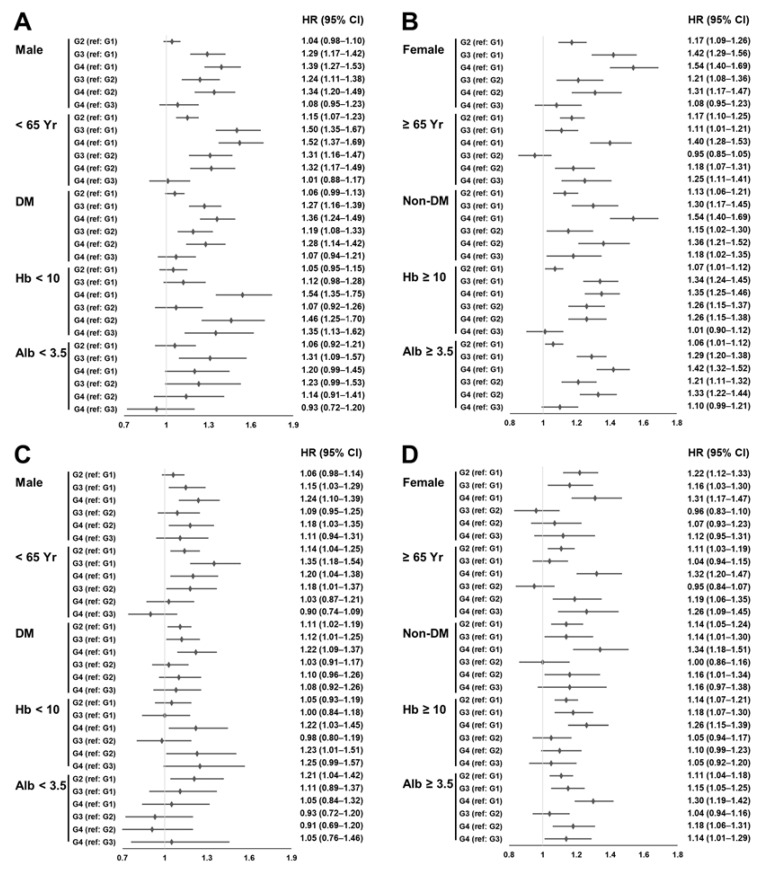
Forest plots of HR and 95% CI by subgroup. (**A**,**B**) Univariate Cox regression analyses. (**C**,**D**) Multivariable Cox regression analyses. Adjusted for sex; Charlson Comorbidity Index score; cause of end-stage renal disease; age; vascular access type; serum creatinine; the duration of hemodialysis; ultrafiltration volume; Kt/V_urea_; serum calcium level; hemoglobin level; serum albumin level; serum phosphorus level; SBP; DBP; the use of anti-hypertensive drugs, aspirin, or statins; erythropoiesis stimulating agent dose; erythropoiesis stimulating agent resistance index; and the use of iron. Abbreviations: Alb < 3.5, serum albumin levels lower than 3.5 g/dL; Alb ≥ 3.5, serum albumin levels equal to or greater than 3.5 g/dL; CI, confidence interval; DBP, diastolic blood pressure; DM, diabetes mellitus; G1, normal iron status; G2, absolute iron deficiency; G3, functional iron deficiency; G4, high iron status; Hb < 10, hemoglobin level < 10 g/dL; Hb ≥ 10, hemoglobin level ≥ 10 g/dL; HR, hazard ratio; SBP, systolic blood pressure; Yr, years.

**Table 1 nutrients-15-02577-t001:** Patient clinical characteristics.

	Group 1 (*n* = 34,539)	Group 2 (*n* = 4476)	Group 3 (*n* = 1719)	Group 4 (*n* = 1656)	*p*
Age (years)	59.8 ± 12.8	58.3 ± 13.2 *	60.9 ± 12.9 *^#^	60.8 ± 12.7 *^#^	<0.001
Sex (male, %)	20,847 (60.4%)	2511 (56.1%)	855 (49.7%)	816 (49.3%)	<0.001
Hemodialysis vintage (months)	40 (61)	39 (60)	39 (57)	51 (71) *^#+^	<0.001
Underlying causes of ESRD					<0.001
Diabetes mellitus	14,664 (42.5%)	1850 (41.3%)	835 (48.6%)	695 (42.0%)	
Hypertension	9236 (26.7%)	1191 (26.6%)	399 (23.2%)	423 (25.5%)	
Glomerulonephritis	3934 (11.4%)	440 (9.8%)	154 (9.0%)	193 (11.7%)	
Others	2818 (8.2%)	423 (9.5%)	154 (9.0%)	162 (9.8%)	
Unknown	3887 (11.3%)	572 (12.8%)	177 (10.3%)	183 (11.1%)	
CCI score	7.1 ± 2.8	7.1 ± 2.8	7.5 ± 2.9 *^#^	7.5 ± 3.0 *^#^	<0.001
Follow-up duration (months)	79 (45)	79 (42) *	71 (46) *^#^	66 (50) *^#^	<0.001
Vascular access type					<0.001
Arteriovenous fistula	29,670 (85.9%)	3719 (83.1%)	1459 (84.9%)	1392 (84.1%)	
Arteriovenous graft	4869 (14.1%)	757 (16.9%)	260 (15.1%)	264 (15.9%)	
Kt/V_urea_	1.52 ± 0.27	1.49 ± 0.28 *	1.54 ± 0.28 ^#^	1.60 ± 0.29 *^#+^	<0.001
Ultrafiltration volume (%/BW/session)	3.92 ± 1.62	3.98 ± 1.63 *	3.84 ± 1.61 *^#^	3.95 ± 1.77 *^#^	<0.001
Hemoglobin (g/dL)	10.7 ± 0.8	10.6 ± 1.0 *	10.4 ± 0.8 *^#^	10.4 ± 0.9 *^#^	<0.001
Serum albumin (g/dL)	3.99 ± 0.34	3.95 ± 0.34 *	3.93 ± 0.36 *	3.91 ± 0.36 *^#^	<0.001
Serum phosphorus (mg/dL)	5.0 ± 1.4	5.2 ± 1.5 *	4.8 ± 1.4 *^#^	4.8 ± 1.5 *^#^	<0.001
Serum calcium (mg/dL)	8.94 ± 0.87	8.88 ± 0.88 *	8.94 ± 0.84	8.98 ± 0.90 ^#^	<0.001
SBP (mmHg)	141 ± 16	142 ± 16 *	142 ± 16	140 ± 16 ^#^	<0.001
DBP (mmHg)	78 ± 9	79 ± 10 *	78 ± 10 ^#^	78 ± 10 ^#^	<0.001
Serum creatinine (mg/dL)	9.7 ± 2.7	9.7 ± 2.9	9.2 ± 2.8 *^#^	8.8 ± 2.5 *^#+^	<0.001
Use of antihypertensive drug	23,456 (67.9%)	2948 (65.9%)	1175 (68.4%)	1106 (66.8%)	0.036
Use of aspirin	15,342 (44.4%)	1958 (43.7%)	787 (45.8%)	705 (42.6%)	0.233
Use of statin	9388 (27.2%)	1230 (27.5%)	561 (32.6%)	427 (25.8%)	<0.001
Transferrin saturation (%)	33 (15)	16 (5) *	17 (4) *^#^	41 (28) *^#+^	<0.001
Serum ferritin (ng/mL)	197 (212)	74 (87) *	310 (177) *^#^	1008 (392) *^#+^	<0.001
Use of iron	21,676 (62.8%)	2458 (54.9%)	1086 (63.2%)	902 (54.5%)	<0.001
ESA dose (IU/week)	5050 (4820)	5470 (6120) *	5740 (5110) *^#^	5510 (5130) *	<0.001
ERI (IU/kg/g/dL)	8.0 (8.0)	8.8 (10.6) *	9.4 (9.0) *^#^	9.5 (9.7) *^#^	<0.001

Categorical variables are presented as numbers (percentages), while continuous variables with a normal distribution are presented as means ± standard deviations, and those without a normal distribution are presented as median (interquartile range). The statistical significance of differences between the means of the variables is evaluated using one-way analysis of variance for continuous variables, followed by Tukey’s post hoc examination, but those without normal distribution were compared using the Kruskal–Wallis test, followed by Dunn’s post hoc test with Bonferroni correction. Pearson’s χ^2^ test or Fisher’s exact test is used for categorical variables. Group 1, patients with normal iron status; Group 2, patients with absolute iron deficiency; Group 3, patients with functional iron deficiency; and Group 4, patients with high iron status. Abbreviations: CCI, Charlson Comorbidity index; DBP, diastolic blood pressure; ESRD, end-stage renal disease; ESA, erythropoiesis-stimulating agent; ERI, ESA resistance index; SBP, systolic blood pressure. * *p* < 0.05 vs. group 1, ^#^
*p* < 0.05 vs. group 2, ^+^
*p* < 0.05 vs. group 3.

**Table 2 nutrients-15-02577-t002:** Patient survival analyzed using Cox regression.

	Univariate Analysis		Multivariable Analysis	
	Hazard Ratio (95% CI)	*p*-Value	Hazard Ratio (95% CI)	*p*-Value
Group (ref: Group 1)				
Group 2	1.08 (1.03–1.13)	<0.001	1.12 (1.06–1.19)	<0.001
Group 3	1.32 (1.24–1.41)	<0.001	1.14 (1.05–1.24)	0.001
Group 4	1.43 (1.34–1.53)	<0.001	1.27 (1.17–1.38)	<0.001
Underlying disease of ESRD (ref: DM)	0.80 (0.79–0.81)	<0.001	0.90 (0.89–0.92)	<0.001
Age	1.06 (1.06–1.07)	<0.001	1.06 (1.06–1.06)	<0.001
Vascular access (ref: AVF)	1.49 (1.43–1.54)	<0.001	1.17 (1.12–1.22)	<0.001
CCI score	1.14 (1.14–1.15)	<0.001	1.07 (1.06–1.08)	<0.001
Sex (ref: male)	0.86 (0.83–0.89)	<0.001	0.71 (0.68–0.74)	<0.001
Hemodialysis vintage (ref: <1229 days)	1.03 (0.99–1.06)	0.079	1.38 (1.32–1.43)	<0.001
UFV (increase per 1% of BW)	0.97 (0.96–0.98)	<0.001	1.05 (1.04–1.06)	<0.001
KtV_urea_	0.90 (0.84–0.95)	<0.001	0.77 (0.71–0.83)	<0.001
Serum creatinine	0.87 (0.86–0.87)	<0.001	0.93 (0.92–0.94)	<0.001
Serum albumin	0.39 (0.37–0.40)	<0.001	0.65 (0.62–0.69)	<0.001
Serum phosphorus	0.86 (0.85–0.87)	<0.001	1.04 (1.02–1.05)	<0.001
Serum calcium	0.94 (0.92–0.95)	<0.001	1.07 (1.04–1.09)	<0.001
Hemoglobin	0.87 (0.85–0.88)	<0.001	0.94 (0.91–0.96)	<0.001
SBP	1.01 (1.01–1.01)	<0.001	1.01 (1.00–1.01)	<0.001
DBP	0.98 (0.98–0.99)	<0.001	1.01 (1.00–1.01)	0.048
Use of anti–hypertensive drug	1.16 (1.12–1.19)	<0.001	1.02 (0.98–1.07)	0.345
Use of aspirin	1.17 (1.13–1.20)	<0.001	1.00 (0.96–1.04)	0.961
Use of statin	1.15 (1.12–1.19)	<0.001	0.99 (0.95–1.03)	0.548
ESA dose (ref: <5130 IU/week)	1.20 (1.17–1.24)	<0.001	0.98 (0.94–1.03)	0.419
ERI	1.02 (1.02–1.02)	<0.001	1.01 (1.01–1.01)	<0.001
Use of iron	0.92 (0.89–0.95)	<0.001	0.97 (0.93–1.01)	0.110

Multivariable analysis is performed with adjustments made for several variables, including sex; age; vascular access type; underlying cause of ESRD; hemodialysis vintage; CCI score; UFV; serum creatinine level; hemoglobin level; Kt/V_urea_; serum albumin level; serum phosphorus level; serum calcium; SBP; DBP; the use of anti–hypertensive drugs, aspirin, or statins; ESA dose; ERI; and use of iron. It was performed using the enter mode and shows a single Cox regression model using multiple independent variables (variables expressed in Table) and a single dependent variable (mortality). Group 1, patients with normal iron status; Group 2, patients with absolute iron deficiency; Group 3, patients with functional iron deficiency; and Group 4, patients with high iron status. HRs are expressed as a change according to the increase per 1 year for age; increase per 1 score for CCI score; increase per 1 unit for Kt/V_urea_; increase per 1 g/dL for hemoglobin or serum albumin; increase per 1 mg/dL for serum creatinine, phosphorus, or calcium; increase per 1 IU/kg/d/gL for ERI; and increase per 1 mmHg for SBP or DBP. Abbreviations: AVF, arteriovenous fistula; BW, body weight; CCI, Charlson Comorbidity Index; CI, confidence interval; DBP, diastolic blood pressure; DM, diabetes mellitus; ESA, erythropoiesis stimulating agent; ESRD, end-stage renal disease; ERI, erythropoiesis stimulating agent resistance index; SBP, systolic blood pressure; UFV, ultrafiltration volume.

## Data Availability

The raw data were generated at the Health Insurance Review and Assessment Service. The database can be requested from the Health Insurance Review and Assessment Service by sending a study proposal including the purpose of this study, study design, and duration of analysis through an e-mail (turtle52@hira.or.kr) or through the website “https://www.hira.or.kr” (assessed on 20 April 2023). The authors cannot distribute the data without permission.

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
