# Peer review of "Association between Iron Status and Survival in Patients on Chronic Hemodialysis"

_nutrients, 2023, doi:10.3390/nu15112577_

Round 1

Reviewer 1 Report

Kang et al. conducted a retrospective observational study to investigate the association between iron status and mortality risk among maintenance hemodialysis (HD) patients. The authors found that HD patients with normal iron status had a better survival rate when compared with those with abnormal iron status. Although this study had a relatively large sample size, the major issue is the lack of novelty, unclear grouping of the patients, and the problems on statistical analyses. Furthermore, the manuscript appears to be not well prepared for the absence of results and Figure 2 indicating subgroup analyses in the text. Survival in patients with a high ferritin level could confound by inflammation drastically, the authors need to solve this problem as well.

Major points:

#1. In this study, the authors classified HD patients into 4 groups as described in line 82 to 87. In this way, two groups of patients with TSAT >20% + ferritin <200 ng/mL and TSAT <20% + ferritin >800 ng/mL cannot be included. In fact, these patients were not excluded from the study. The authors should make this clearer and re-analyze the data.

#2. Certain results in univariate and multivariate Cox regression analyses for survival are completely opposite, and this is difficult to understand and is also confusing. For example, higher ultrafiltration volume is protective for survival in univariate analysis, but is detrimental in multivariate Cox regression. Similarly, serum phosphorous level, serum calcium level, and pre-HD diastolic blood pressure level appear contradictory in Cox model. Besides, it could be more meaningful in clinical settings for ultrafiltration with regard to representing the percentage of weight, instead of the absolute volume.

#3. It is unclear about the significance of incremental per one day of HD vintage for survival. The authors should consider performing the sensitivity test or dichotomize this variable. Similarly, what is the clinical significance of increase per 1 IU/week in ESA dose? It is not very useful.

#4. The authors should add the number of patients at risk in Figure 1. And it is unclear about this figure for illustrating the survival rate during follow-up period or for 5-year survival rate as described in line 150. A p-value using the log-rank test should be showed clearly in the text and in the figure.

#5. Malnutrition and inflammation are ideally added in the subgroup analyses. As mentioned above, the results and figure 2 indicating subgroup analyses is not seen in the text.

#6. The authors should add the information of iron treatment, ESA doses, and ESA resistance index in Table 1.

#7. Were TSAT and serum ferritin collected from the patients monthly? It appears inconsistent between the descriptions in line 76−81 and in line 255-257.

#8. In the introduction, the authors should have depicted more clearly the relationship between iron and the risk of death.

Minor points:

#1. The patient number in group 2 is incorrect in the abstract.

#2. There are some typos in the text, such as “stain” in table 1 and in line 138.

#3. It could be better to generate a figure illustrating the conditions at the follow-up end-point in each group of patients (in line 144 to 149).

Moderate editing of English language is required

Author Response

Major points:

#1. In this study, the authors classified HD patients into 4 groups as described in line 82 to 87. In this way, two groups of patients with TSAT >20% + ferritin <200 ng/mL and TSAT <20% + ferritin >800 ng/mL cannot be included. In fact, these patients were not excluded from the study. The authors should make this clearer and re-analyze the data.

Answer: Thank you for your comments. There was some confusions in the definition of the groups. We have revised the definition of the groups as follows: Group 1 (normal iron status), TSAT > 20% and serum ferritin ≤ 800 ng/mL; Group 2 (absolute iron deficiency), TSAT ≤ 20% and serum ferritin ≤ 200 ng/mL; Group 3 (functional iron deficiency), TSAT ≤ 20% and serum ferritin 200–800 ng/mL; and Group 4 (high iron status), serum ferritin 800 ng/mL.

#2. Certain results in univariate and multivariate Cox regression analyses for survival are completely opposite, and this is difficult to understand and is also confusing. For example, higher ultrafiltration volume is protective for survival in univariate analysis, but is detrimental in multivariate Cox regression. Similarly, serum phosphorous level, serum calcium level, and pre-HD diastolic blood pressure level appear contradictory in Cox model. Besides, it could be more meaningful in clinical settings for ultrafiltration with regard to representing the percentage of weight, instead of the absolute volume.

Answer: Thank you for your comments. We have revised ultrafiltration volume (kg/session) to ultrafiltration volume (percent per post-HD body weight). Further, Table 2 has been amended using the revised variables as follows:

Table 2. Cox regression analyses for patient survival

Univariate

Multivariate

HR (95% CI)

P

HR (95% CI)

P

Group

  Ref: Group 1

    Group 2

1.08 (1.03–1.13)

<0.001

1.13 (1.071.20)

<0.001

Group 3

1.32 (1.24–1.41)

<0.001

1.16 (1.071.26)

<0.001

Group 4

1.43 (1.34–1.53)

<0.001

1.29 (1.191.40)

<0.001

  Ref: Group 2

    Group 3

1.22 (1.13–1.32)

<0.001

1.03 (0.931.13)

0.580

    Group 4

1.32 (1.22–1.43)

<0.001

1.13 (1.021.25)

0.014

  Ref: Group 3

    Group 4

1.08 (0.98–1.19)

0.102

1.10 (0.981.24)

0.089

Underlying disease of ESRD (ref: DM)

0.80 (0.79–0.81)

<0.001

0.91 (0.890.92)

<0.001

Age (increase per 1 year)

1.06 (1.06–1.07)

<0.001

1.06 (1.061.06)

<0.001

Type of vascular access (ref: arteriovenous fistula)

1.49 (1.43–1.54)

<0.001

1.17 (1.121.22)

<0.001

CCI score (increase per 1 score)

1.14 (1.14–1.15)

<0.001

1.07 (1.061.08)

<0.001

Sex (ref: male)

0.86 (0.83–0.89)

<0.001

0.72 (0.690.75)

<0.001

Hemodialysis vintage (ref: <1229 days)

1.03 (0.99-1.06)

0.079

1.38 (1.331.44)

<0.001

Ultrafiltration volume (increase per 1% of BW)

0.97 (0.96-0.98)

<0.001

1.05 (1.041.06)

<0.001

KtVurea (increase per 1 unit)

0.90 (0.84–0.95)

<0.001

0.79 (0.730.85)

<0.001

Hemoglobin (increase per 1 g/dL)

0.87 (0.85–0.88)

<0.001

0.91 (0.880.93)

<0.001

Serum albumin (increase per 1 g/dL)

0.39 (0.37–0.40)

<0.001

0.64 (0.610.68)

<0.001

Serum creatinine (increase per 1 mg/dL)

0.87 (0.86–0.87)

<0.001

0.93 (0.920.94)

<0.001

Serum phosphorus (increase per 1 mg/dL)

0.86 (0.85–0.87)

<0.001

1.04 (1.021.05)

<0.001

Serum calcium (increase per 1 mg/dL)

0.94 (0.92–0.95)

<0.001

1.07 (1.041.09)

<0.001

Systolic blood pressure (increase per 1 mmHg)

1.01 (1.01–1.01)

<0.001

1.01 (1.001.01)

<0.001

Diastolic blood pressure (increase per 1 mmHg)

0.98 (0.98–0.99)

<0.001

1.01 (1.001.01)

0.045

Use of anti–hypertensive drug

1.16 (1.12–1.19)

<0.001

1.01 (0.971.06)

0.524

Use of aspirin

1.17 (1.13–1.20)

<0.001

1.00 (0.961.03)

0.855

Use of statin

1.15 (1.12–1.19)

<0.001

0.98 (0.951.02)

0.338

ESA dose (ref: <5130 IU/week)

1.20 (1.17–1.24)

<0.001

1.08 (1.041.12)

<0.001

Use of iron

0.92 (0.89–0.95)

<0.001

0.97 (0.931.01)

0.102

Multivariate analysis was adjusted for age; sex; underlying cause of ESRD; CCI score; vascular access type; hemodialysis vintage; ultrafiltration volume; Kt/Vurea; hemoglobin level; serum albumin level; serum creatinine level; serum phosphorus level; serum calcium level; systolic blood pressure; diastolic blood pressure; use of anti-hypertensive drugs, aspirin, and statins; ESA dose; and use of iron, and was performed using enter mode. Group 1, patients with normal iron status; Group 2, patients with absolute iron deficiency; Group 3, patients with functional iron deficiency; and Group 4, patients with high iron status.

Abbreviations: BW, body weight; CCI, Charlson comorbidity index; CI, confidence interval; DM, diabetes mellitus; ESA, erythropoiesis stimulating agent; ESRD, end-stage renal disease; HR, hazard ratio.

Increases in ultrafiltration volume, serum calcium/phosphorus levels, and diastolic blood pressure are well-known risk factor for mortality in patients undergoing HD. However, the revised results also showed opposite outcomes for ultrafiltration volume, serum phosphorus and serum calcium levels, and diastolic blood pressure between univariate and multivariate analyses. These results may be associated with the confounding effects of nutritional status. Good nutritional intake may lead to high calcium and phosphorus levels, and ultrafiltration volume and diastolic blood pressure also can increase via volume repletion. These confounding effects of nutritional status may be attenuated by multivariable analysis, and the hazard ratio can recover the values with an independent effect. Our results follow this trend similarly. We have added these comments in the Results and Discussion sections.

#3. It is unclear about the significance of incremental per one day of HD vintage for survival. The authors should consider performing the sensitivity test or dichotomize this variable. Similarly, what is the clinical significance of increase per 1 IU/week in ESA dose? It is not very useful.

Answer: Thank you for your comments. As the reviewer pointed out, we have revised HD vintage and ESA dose values as continuous variables to dichotomous variables using median values as follows: HD vintage, short group with HD < 1229 days and long group with HD ≥ 1229 days; ESA dose, low dose with < 5130 IU/week and high dose with ≥ 5130 IU/week. We have revised the results of multivariate Cox regression analyses using these variables. Detailed corrections are described in the answers of previous comments (#2).

#4. The authors should add the number of patients at risk in Figure 1. And it is unclear about this figure for illustrating the survival rate during follow-up period or for 5-year survival rate as described in line 150. A p-value using the log-rank test should be showed clearly in the text and in the figure.

Answer: Thank you for your comments. We have revised the Figure and legend as follows:

Figure 2. Kaplan–Meier curves for patient survival per study group. The 5-year survival rates per group are expressed as dotted lines and P-values using log-rank test are added in the lower left corner of the graph.

Abbreviations: Group 1, patients with normal iron status; Group 2, patients with absolute iron deficiency; Group 3, patients with functional iron deficiency; Group 4, patients with high iron status

Kaplan–Meier curves showed that Group 1 had the highest patient survival rate among the four groups, followed by Group 2. The patient survival rate did not differ significantly between Groups 3 and 4. We have added the revised Figure and comments in the Results section.

#5. Malnutrition and inflammation are ideally added in the subgroup analyses. As mentioned above, the results and figure 2 indicating subgroup analyses is not seen in the text.

Answer; Thank you for your comments. We have revised Figure 3 with subgroups for serum albumin level.

Figure 3. Forest plots of HR and 95% CI by subgroup. (A and B) Univariate Cox regression analyses. (C and D) Multivariate Cox regression analyses.

Multivariate Cox regression analyses showed that subgroup analyses using sex, age, presence of DM, hemoglobin level ≥ 10 g/dL, or serum albumin level ≥ 3.5 g/dL had similar trends with those using the total cohort. However, subgroup analyses using patients with hemoglobin level < 10 g/dL or serum albumin level < 3.5 g/dL showed a weak statistical significant difference than those with hemoglobin level ≥ 10 g/dL, or serum albumin level ≥ 3.5 g/dL.

Our subgroup analyses showed a weak association between iron status and patient survival in patient with anemia or low serum albumin levels. This finding may be associated with unresolved confounding factors. Herein, anemic patients with normal iron status and those with iron deficiency had similar patient survival rates. Anemia in patients with normal iron status could occur due to other factors, such as inadequate ESA dose, hyperparathyroidism, or inflammation, which are associated with poor outcomes [3]. Non-adjustment for these factors may be associated with similar patient survival rates between patients with normal iron and patients with iron deficiency in subgroup analyses using patients with anemia. Further, subgroup analyses using patients with low serum albumin levels were also similar to those using patients with anemia. In our study, Group 4 included patients with absolute iron overload or high ferritin levels due to inflammation. Serum albumin is a well-known acute phase reactant and inversely correlated with inflammation. In subgroup analyses using patients with low serum albumin, the weak association between high iron status and patient survival would be associated with the confounding effect of inflammation or malnutrition. However, normal serum albumin levels can rule out a high inflammatory condition. Subgroup analyses using patients with normal serum albumin levels were similar to those using the total cohort and may thus aid in concluding that abnormal iron status is an independent risk factor for mortality in HD patients without high inflammation status.

We have added these comments in the Results and Discussion sections.

References

[3] KDIGO Anemia Work Group. KDIGO clinical practice guideline for anemia in chronic kidney disease. Kidney Int Suppl. 2012, 2:279-335.

#6. The authors should add the information of iron treatment, ESA doses, and ESA resistance index in Table 1.

Answer: Thank you for your comments. We have added these data in the Table 1.

#7. Were TSAT and serum ferritin collected from the patients monthly? It appears inconsistent between the descriptions in line 76−81 and in line 255-257.

Answer; Thank you for your comments. There seems to be a confusion regarding the interval of iron status evaluation. Despite previous guidelines, HIRA recommends that iron status be evaluated monthly throughout the 6 months of the HD Quality Assessment Program. TSAT (%) and serum ferritin levels were measured monthly over 6 months. We used averaged data for TSAT and ferritin levels. We have added these comments in the Methods section.

#8. In the introduction, the authors should have depicted more clearly the relationship between iron and the risk of death.

Answer: Thank you for your comments.

Iron is an essential element in most human cells and a cofactor of vital proteins and enzymes [1]. Iron dysregulation is common in patients undergoing HD. Anemia in HD patients is caused by two important factors: insufficient erythropoietin and abnormal iron status. Iron deficiency in anemia can be caused by absolute or functional iron deficiency. Thus, sustaining a normal iron balance is essential for maintaining normal hemoglobin levels. Beside anemia, iron plays an important role in maintaining mitochondria respiration in cardiomyocytes. Therefore, iron deficiency increases morbidity and mortality in patients undergoing HD. However, iron overload is also associated with high mortality in patients on HD because of the increased infection risk due to an abnormal immune response and cardiovascular complications caused by endothelial dysfunction or myocardial iron deposit [2]. Consequently, maintaining a normal iron status assists in improving patient survival.

We have added these comments in the Introduction section.

Added references

[1] Xie Y, Liu F, Zhang X, Jin Y, Li Q, Shen H, Fu H, Mao J. Benefits and risks of essential trace elements in chronic kidney disease: a narrative review. Ann Transl Med. 2022 Dec;10(24):1400.

[2] Walther CP, Triozzi JL, Deswal A. Iron deficiency and iron therapy in heart failure and chronic kidney disease. Curr Opin Nephrol Hypertens. 2020 Sep;29(5):508-514. 

Minor points:

#1. The patient number in group 2 is incorrect in the abstract.

Answer; Thank you for your comments. We have corrected the error.

#2. There are some typos in the text, such as “stain” in table 1 and in line 138.

Answer; Thank you for your comments. We have corrected the error.

#3. It could be better to generate a figure illustrating the conditions at the follow-up end-point in each group of patients (in line 144 to 149).

Answer; Thank you for your comments. We have added Figure 1 with patient status at the endpoint of follow-up as follows:

Figure 1. Patients’ status at the endpoint of follow-up.

Reviewer 2 Report

Dear authors,

I congratulate you for this well conducted, methodogically sound and scientifically interesting study. 

Below are several minor changes suggested, other than those, the article is ready for publication as is.

Line 46 replace "some" with "several studies"

Line 52-3 "proper interventions based on the the type of iron status abnormality"

Line 136 "diabetes mellitus"

Author Response

Dear authors,

I congratulate you for this well conducted, methodogically sound and scientifically interesting study. 

Below are several minor changes suggested, other than those, the article is ready for publication as is.

Line 46 replace "some" with "several studies"

Line 52-3 "proper interventions based on the the type of iron status abnormality"

Line 136 "diabetes mellitus"

Answer; Thank you for your comments. We have corrected the typographical errors. We look forward to hearing from you and would be happy to make further changes, if required. Thank you once again.

Round 2

Reviewer 1 Report

The authors properly clarify most of the concerning issues in their revised manuscript. The novelty of the study improves after well-performed subgroup analyses, indicating that abnormal iron status had a greater influence on survival among HD patients with relatively better nutritional status or less inflammation. Based on these new findings, the authors are encouraged to revise the abstract and the section of conclusions. Furthermore, certain problems in the text and statistical analyses need to be fixed by the authors. Please consider the following comments.

1. Please revise the abstract and the conclusions in the text based on the important findings in the subgroup analyses.

2. The authors use the median value of HD vintage (1129 days) and ESA dose (5130 IU/week) to generate dichotomous variables in Cox regression analyses. Please add these details in the section of statistical analyses in the revised manuscript.

3. Please consider add ESA resistance index (ERI) as a variable in Cox regression analyses in Table 2 and Figure 3.

4. In my understanding, the p-value in Figure 2 appears to be long-rank p-value, but not p-value for trend (in line 163). Besides, in line 164, the authors described p=0.001 for Group 2 vs. Group 3 or 4, which is not consistent with p-value showed in Figure 2.

5. The descriptions in line 216 to 221 are totally unclear. Why the authors performed “multivariate analyses” for ESA dose and ERI?

6. The definition of patients in group 4 (high iron status) is incorrect. (Line 96)

7. It is quite interesting that more than half of patients with high iron status received treatment of iron. The authors are encouraged to explain the reasons.

8. In subgroup analyses, the associations between iron status and survival appear to be different among patients <65 and 65 years of age. The authors may consider address this finding and possible explanations in the text.

Minor editing of English language required

Author Response

Our point-by-point response to the reviewer’s comments and suggestions is listed below: We thank you for taking the time and effort necessary to review our manuscript and provide us with these valuable comments and suggestions. Accordingly, we revised our manuscript and made changes to it.

The authors properly clarify most of the concerning issues in their revised manuscript. The novelty of the study improves after well-performed subgroup analyses, indicating that abnormal iron status had a greater influence on survival among HD patients with relatively better nutritional status or less inflammation. Based on these new findings, the authors are encouraged to revise the abstract and the section of conclusions. Furthermore, certain problems in the text and statistical analyses need to be fixed by the authors. Please consider the following comments.

  1. Please revise the abstract and the conclusions in the text based on the important findings in the subgroup analyses.

 Answer: We thank you for your comments. We have added some comments to the Abstract and Conclusion sections on the subgroup analyses.

We revised the abstract section as follows including the following changes in italic font:

The aim of the study was to evaluate survival rates according to iron status in patients undergoing maintenance hemodialysis (HD). Thus, the National HD Quality Assessment Program dataset and claims data were used for analysis (n = 42,390). The patients were divided into four groups according to transferrin saturation rate and serum ferritin levels: Group 1 (n = 34,539, normal iron status); Group 2 (n = 4,476, absolute iron deficiency); Group 3 (n = 1,719, functional iron deficiency); and Group 4 (n = 1,656, high iron status). Using univariate and multivariate analyses, Group 1 outperformed the other three groups regarding patient survival. Using univariate analysis, although Group 2 showed a favorable trend in patient survival rates compared with Groups 3 and 4, the statistical significance was weak. Group 3 exhibited similar patient survival rates to Group 4. Using multivariate Cox regression analysis, Group 2 had similar patient survival rates to Group 3. Subgroup analyses according to sex, diabetic status, hemoglobin level ≥ 10 g/dL, and serum albumin levels ≥ 3.5 g/dL indicated similar trends to those by the total cohort. However, subgroup analysis based on patients with hemoglobin level < 10 g/dL or serum albumin levels < 3.5 g/dL showed a weak statistical significant difference compared with those with hemoglobin level ≥ 10 g/dL, or serum albumin levels ≥ 3.5 g/dL. In addition, the survival difference between Group 4 and other groups was greater in old patients than in young ones. Patients with normal iron status exhibited the highest survival rates. Patient survival rates were similar or differed only modestly among the groups with abnormal iron status. In addition, most subgroup analyses revealed similar trends to those according to the total cohort. However, subgroup analyses based on age, hemoglobin, or serum albumin levels showed a different trend.

  1. The authors use the median value of HD vintage (1129 days) and ESA dose (5130 IU/week) to generate dichotomous variables in Cox regression analyses. Please add these details in the section of statistical analyses in the revised manuscript.

 Answer: Thank you for your suggestions. We have added some comments to the Statistical analyses section as follows:

In addition, we used categorical data based on median values for HD vintage and ESA dose. The median values were 1229 days for HD vintage and 5130 IU/week for ESA dose. The two variables were adjusted for as dichotomous variables: HD vintage, short group with HD < 1229 days and long group with HD ≥ 1229 days; ESA dose, low dose with < 5130 IU/week and high dose with ≥ 5130 IU/week.

  1. Please consider add ESA resistance index (ERI) as a variable in Cox regression analyses in Table 2 and Figure 3.

Answer: Thank you for your suggestions. Accordingly, we have added “ERI” to a covariate for multivariate Cox regression analyses. In addition, we have revised the results in Table 2 and Figure 3. The revised results showed similar trend to those without ERI.

  1. In my understanding, the p-value in Figure 2 appears to be long-rank p-value, but not p-value for trend (in line 163). Besides, in line 164, the authors described p=0.001 for Group 2 vs. Group 3 or 4, which is not consistent with p-value showed in Figure 2.

 Answer: We thank you for your comments. We have indicated the P-values in the lower left corner of the graph to P-values for pairwise comparison with log-rank test and added P-values of the trend to the Figure. In addition, we have replaced “P = 0.001” with “P < 0.001” throughout the manuscript.

  1. The descriptions in line 216 to 221 are totally unclear. Why the authors performed “multivariate analyses” for ESA dose and ERI?

 Answer: Thank you for your comments. We have revised the relevant sentences as follows: The ERI for Groups 1, 2, 3, and 4 were 8.7 ± 6.8, 9.7 ± 9.0, 10.7 ± 8.7, and 10.9 ± 9.9 IU/kg/g/dL, respectively (P < 0.001 for trend). We performed pairwise comparison between two groups and showed that ERI was significantly lower in Group 1 than Groups 2, 3, or 4 (P < 0.05). The ERI value in Group 2 was significantly lower than those in Groups 3 or 4 (P < 0.05). There were significant differences in baseline characteristics and we performed multivariate analyses to identify independent effect for ERI or ESA dose of iron status. Multivariate analyses showed that the mean ESA doses in Groups 1, 2, 3, and 4 were 5,396, 5,758, 5,666, and 5,702 IU/week, respectively (P < 0.001 for trend). The mean ESA dose was significantly lower in Group 1 than Group 2 (P < 0.05). The mean ERI for Groups 1, 2, 3, and 4 was 8.9, 9.7, 9.6, and 9.9 IU/kg/g/dL, respectively (P < 0.001 for trend). The mean ERI in Group 1 was significantly lower than those in Groups 2, 3, or 4 (P < 0.05). Analysis of covariance was used to adjust baseline characteristics.

We have added these comments in the Methods and Results sections.

  1. The definition of patients in group 4 (high iron status) is incorrect. (Line 96)

Answer: We thank you for pointing this out. We have revised the definition of Group 4 to “800 ng/mL < serum ferritin”. In addition, we have added “All patients with 800 ng/mL < serum ferritin had 20% < TSAT.” to the Methods section.

  1. It is quite interesting that more than half of patients with high iron status received treatment of iron. The authors are encouraged to explain the reasons.

 Answer: Thank you for your suggestions. There may be several reasons for the prescription of iron supplements despite the high iron status. First, some patients in Group 4 would have been prone to inflammatory status and had inadequate response to ESA for managing anemia. These may lead to prescription of iron despite high iron status. Second, iron supplements might have been prescribed without careful consideration of a patient’s iron levels, potentially leading to indiscriminate use. Another factor that could have contributed to this issue would be the patient’s mistaken belief that iron supplements were helpful regardless of their own iron status. Although the issue is beyond the scope of our study, further studies of iron supplementation in these patients would be helpful to attenuate iron overload and its related complications. We have these comments to the Discussion section.

  1. In subgroup analyses, the associations between iron status and survival appear to be different among patients <65 and ≥65 years of age. The authors may consider address this finding and possible explanations in the text.

Answer: We thank you for your suggestion. As you pointed out, survival difference between Group 4 and the other groups was greater in patients ≥ 65 years than in those < 65 years. These may be associated with greater vulnerability to iron toxicity in patients ≥ 65 years than in those < 65 years. High iron status could lead to cell damage by producing hydroxyl free radicals [1]. The older population typically experiences a decrease in defense mechanisms, particularly those related to oxidative stress, which makes them more vulnerable to the impacts results from high levels of iron. High mortality in Group 4 in patients ≥ 65 years would be associated with these issues. We have added these comments to the Results and Discussion sections.

Added reference

[1] Tian Y, Tian Y, Yuan Z, Zeng Y, Wang S, Fan X, Yang D, Yang M. Iron Metabolism in Aging and Age-Related Diseases. Int J Mol Sci. 2022 Mar 25;23(7):3612.

Comments on the Quality of English Language

Minor editing of English language required

Answer: We thank you for your suggestions. We have requested English editing by two native speakers.
